# Microalgae as an Alternative Mineral Source in Poultry Nutrition

**DOI:** 10.3390/vetsci11010044

**Published:** 2024-01-20

**Authors:** Mónica M. Costa, Maria P. Spínola, José A. M. Prates

**Affiliations:** 1CIISA—Centro de Investigação Interdisciplinar em Sanidade Animal, Faculdade de Medicina Veterinária, Universidade de Lisboa, Av. da Universidade Técnica, 1300-477 Lisboa, Portugal; monicacosta@fmv.ulisboa.pt (M.M.C.); mariaspinola@fmv.ulisboa.pt (M.P.S.); 2Laboratório Associado para Ciência Animal e Veterinária (AL4AnimalS), 1300-477 Lisboa, Portugal

**Keywords:** microalga, ash, mineral composition, poultry nutrition, feed

## Abstract

**Simple Summary:**

This review looks at how microalgae can be used as a new and environmentally friendly way to feed chickens. Usually, chickens are given minerals like calcium and iron from sources that can harm the environment and be costly. Microalgae offer a better solution as they are rich in these important minerals and can be grown sustainably, using less land and water. We studied different kinds of microalgae to see which ones have the best nutrients for chickens and how easily these nutrients can be absorbed. We also explored the costs and challenges of using microalgae in chicken feed on a large scale. Our review shows that while microalgae are a promising alternative, there are still economic and safety issues to solve before they can be widely used. The use of microalgae in chicken feed could lead to healthier chickens and a smaller environmental footprint, making this a valuable approach for sustainable farming and food production.

**Abstract:**

This review explores the potential of microalgae as a sustainable and nutritionally rich alternative for mineral supplementation in poultry diets, addressing both the opportunities and challenges in this emerging field. Poultry nutrition, pivotal to the health and productivity of birds, traditionally relies on inorganic and organic mineral sources which, while effective, raise environmental and economic concerns. Microalgae offer a promising solution with their high contents of essential minerals, proteins, vitamins, and bioactive compounds. This review delves into the nutritional profiles of various microalgae, highlighting their rich contents of minerals which are crucial for physiological processes in poultry. It examines the bioavailability of these minerals and their impact on poultry health and productivity. Furthermore, it evaluates the environmental sustainability of microalgae cultivation and acknowledges the challenges in using microalgae in poultry diets, particularly in terms of the economic viability of large-scale production and the consistency of nutrient composition. It discusses the importance of rigorous safety assessments and regulatory compliance, given the potential risks of toxins and heavy metals. Overall, this analysis aims to provide a clear understanding of the role microalgae could play in poultry nutrition and address sustainability challenges in animal agriculture while also considering future perspectives and advancements needed in this field.

## 1. Introduction

Poultry nutrition is a critical aspect of modern animal husbandry, impacting not only the health and growth of the birds but also the quality of the products obtained from them, such as meat and eggs. Essential minerals such as calcium, phosphorus, and potassium and trace elements like iron, zinc, and selenium play pivotal roles in various physiological processes in poultry, including bone development, eggshell formation, oxygen transport in the blood, electrolyte balance, enzyme function, antioxidant defence systems, and immune response [1,2].

The traditional approach to providing minerals in poultry nutrition involves both inorganic and organic sources. Inorganic sources, such as calcium carbonate for calcium and salts like sodium selenite for selenium, are effective but raise environmental concerns due to their extraction and processing methods [3]. Organic sources, including chelated forms of zinc, copper, and manganese, offer better bioavailability but come with higher costs [4]. The extraction of inorganic minerals often involves mining, which can lead to environmental degradation, while the production of organic minerals, though more sustainable, is not entirely eco-friendly. The cost factor is significant, especially for chelated minerals, which can substantially increase feed costs [5]. Additionally, the variable bioavailability of inorganic minerals can lead to economic losses and environmental concerns due to the excretion of unabsorbed minerals [6].

Given these challenges, there is growing interest in sustainable and cost-effective alternatives like microalgae. Microalgae, such as Spirulina (*Arthrospira*) and *Chlorella*, offer a rich mineral profile with enhanced bioavailability and a lower environmental impact compared to traditional methods. Their cultivation can be more sustainable, utilizing resources like wastewater and potentially improving feed efficiency in poultry diets [7,8].

However, the utilization of microalgae in poultry diets is not without challenges. Factors such as the bioavailability of minerals from microalgae, the impact on poultry health and productivity, and the economic viability of incorporating microalgae into feed at a commercial scale are crucial considerations [9]. Therefore, this review aims to explore the potential of microalgae as a source of essential minerals in poultry feeding. It covers various aspects including the nutritional profile of microalgae, the bioavailability of microalgal minerals, impacts on poultry health and productivity, environmental and economic considerations, and the current challenges and future perspectives in this field. This comprehensive analysis seeks to provide a clear understanding of the role microalgae could play in poultry nutrition and to address sustainability challenges in animal agriculture.

## 2. An overview of Microalgae

Microalgae are a diverse group of unicellular photosynthetic organisms found in various aquatic environments and classified into diatoms (*Bacillariophyceae*), green algae (*Chlorophyceae*), golden algae (*Chrysophyceae*), and blue-green algae cyanobacteria (*Cyanophyceae*). The most important phototrophic species belong to the *Arthrospira*, *Chlorella*, *Isochrysis*, and *Porphyridium* genera. Regarding heterotrophic marine organisms, *Schizochytrium*, *Crypthecodinium*, and *Ulkenia* have been cultivated for *n*-3 long-chain polyunsaturated fatty acid (*n*-3 LCPUFA) production. Microalgae are known for their rapid growth and ability to thrive in a range of conditions, including extreme environments [10]. Microalgae are distinct from macroalgae, commonly known as seaweeds, in size and habitat diversity.

The nutritional profile of microalgae is remarkable, often described as a rich source of proteins, lipids, vitamins, and essential minerals. The mineral content in microalgae includes calcium, magnesium, phosphorus, potassium, and trace elements such as iron, zinc, and selenium, which are crucial for animal health [11]. This rich composition makes them a potential alternative to conventional mineral sources in animal nutrition.

Microalgae such as *Spirulina* and *Chlorella* have been extensively studied for their nutritional benefits. *Spirulina*, for example, is renowned for its high protein content and comprehensive profile of essential amino acids and vitamins, while *Chlorella* is valued for its lipid profile, including *n*-3 LCPUFAs [12].

The cultivation of microalgae can be tailored to enhance specific nutritional components, a process known as biofortification. Factors like light intensity, nutrient availability, and salinity can influence the nutritional composition of microalgae [13]. This adaptability allows for the production of microalgae biomass with optimized nutrient profiles for specific applications, such as poultry nutrition.

Environmental sustainability is a significant advantage of microalgae cultivation. Microalgae can be cultivated on non-arable land, using saline or wastewater, and they have a high carbon dioxide fixation rate, contributing to carbon sequestration [14]. These factors position microalgae as a sustainable alternative to traditional agricultural practices for feed production.

Despite these advantages, the commercial application of microalgae in poultry feed faces challenges, including the cost-effectiveness of large-scale production and the consistency of the nutrient composition in the biomass. Moreover, the digestibility and bioavailability of nutrients from microalgae in poultry need to be thoroughly evaluated [15,16]. Previous studies assessed the influence of mechanical and enzymatic pre-treatments on disrupting microalgal cell walls and thus increasing the extraction of algal nutrients, with particular emphasis on *Arthrospira platensis* and *Chlorella vulgaris*. For instance, pre-treating an *A. platensis* biomass with bead milling before in vitro digestion improved protein digestibility by 4% [17]. Moreover, in a recent report, an extrusion pre-treatment was shown to decrease the total protein content released from *A. platensis* into the supernatant due to a reduction in protein solubility, which was suggested to enhance protein bioaccessibility [18]. Regarding the use of enzymatic treatments, Coelho et al. [19] reported a partial degradation of the *A. platensis* cell wall with a consequent extraction of some fatty acids and chlorophyll a after treating the microalga suspension with a mixture of lysozyme and α-amylase. Other studies, using a combination of pepsin and pancreatin for the in vitro digestion of *A. platensis*, showed high dry weight (94.3%) [20], organic matter (86.0%) and protein (81.0%) [21] digestibility. The use of a mechanical pre-treatment (i.e., extrusion) followed by an enzymatic (i.e., pancreatin) pre-treatment was recently related to the extraction and hydrolysis of 18 to 26 kDa protein fractions (phycocyanin subunits) [22]. Considering the pre-treatments applied to *C. vulgaris*, high-pressure homogenization, sonication, or ball milling could increase lipid [23] and carotenoid [24] bioaccessibility and crude protein digestibility [17], respectively. Similar pre-treatments were also shown to promote protein diffusion from a *C. vulgaris* biomass into an algal supernatant [25]. Recently, bead milling or microwave treatments were demonstrated to cause an increase in the extraction of high-molecular-weight (66 to 96 kDa) protein fractions from a *C. vulgaris* biomass, whereas the extrusion method enhanced the release of total peptides [26]. In addition, a four-carbohydrase mixture led to the partial disruption of the *C. vulgaris* cell wall, followed by a release of total protein, carotenoids, and some fatty acids [27]. In another in vitro study, a pepsin and pancreatin mixture led to high protein digestibility in *C. vulgaris* (up to 76%) [21]. This benefit of using pancreatin was also demonstrated by Kose et al. [28]. Overall, attempts have been made to increase nutrient bioaccessibility from microalgal biomass, although these efforts did not encompass an evaluation of effects on mineral extraction from pre-treated microalgae.

## 3. Mineral Composition of Microalgae

The mineral composition of microalgae is a critical aspect that enhances their value as a potential component in poultry feed thanks to their rich and varied nutritional profile. Microalgae are distinguished by their high contents of essential minerals, along with proteins, vitamins, and bioactive compounds, all of which play vital roles in poultry health and development. A comprehensive analysis of the main microalgae species used in animal feed, presented in Table 1, highlights this mineral diversity. The species analysed include *Arthrospira* sp., *Chlorella* sp., *Isochrysis* sp., *Porphyridium* sp., and *Schizochytrium* sp. The mineral content, measured on a dry weight basis, covers a range of the main macrominerals (calcium, potassium, magnesium, sodium, phosphorus, and sulphur) and microminerals (copper, iron, manganese, and zinc).

*Arthrospira* sp. shows a wide range in ash content from 6.10% to 34.8%, averaging at 9.87%. Its calcium levels vary between 0.23 and 10.3 g/kg, and it also exhibits a broad range of iron content from 106 to 1036 mg/kg. Additionally, *Arthrospira* sp. contains significant amounts of potassium, ranging from 10.9 to 29.1 g/kg, alongside notable quantities of magnesium, manganese, sodium, phosphorus, and zinc. *Chlorella* sp. also demonstrates a considerable variation in ash content, ranging from 5.50% to 27.3% with an average of 10.7%. Its calcium content fluctuates between 0.36 and 53.3 g/kg, and its iron content spans a wide spectrum from 187 to 5,400 mg/kg. *Chlorella* sp. is rich in other essential minerals too, such as potassium, magnesium, manganese, and sodium, which contribute to its nutritional value. *Isochrysis* sp. and *Porphyridium* sp. are distinguished by even higher average ash contents of 18.7% and 23.1%, respectively, indicating their robust mineral profiles. Particularly noteworthy is *Porphyridium* sp., which exhibits the highest iron content among the analysed species, reaching up to 11,101 mg/kg. Finally, *Schizochytrium* sp., while presenting the lowest range in ash content (from 3.81% to 10.0%, average 7.37%), still contributes significantly to overall mineral diversity. It is characterized by essential nutrients including calcium, potassium, and phosphorus, albeit in varying concentrations.

The mineral contents in various microalgae species exhibit significant variation, which is key to understanding their nutritional potential in poultry diets. The ash content, indicative of the total mineral presence, varies widely among species such as *Arthrospira* sp., *Chlorella* sp., *Isochrysis* sp., *Porphyridium* sp., and *Schizochytrium* sp. This variation in ash content reflects the rich mineral makeup of these microalgae. Particularly, *Porphyridium* sp. stands out with one of the highest percentages of ash content, emphasizing its dense mineral composition. In terms of calcium and iron, there is notable variability across these species. *Arthrospira* sp. and *Chlorella* sp., for instance, display wide ranges in their calcium contents, suggesting their potential value in poultry diets that require these minerals. Additionally, *Porphyridium* sp. is distinguished by its exceptionally high iron content, which could be crucial for addressing iron deficiencies in poultry feed. The potassium contents in these microalgae also show significant variation, which is important considering the role of potassium in several physiological functions in poultry. Alongside these minerals, microalgae species contain varied levels of other essential minerals such as magnesium, manganese, sodium, phosphorus, and zinc. These minerals are essential for various aspects of poultry health, including bone development, enzyme functions, and immune response.

Overall, the diverse range of mineral content in these microalgae species highlights their potential as versatile and rich sources of essential nutrients for poultry, underscoring the possibility of their use in enhancing poultry diets. This variability allows for potential customization and the targeting of specific nutritional needs in poultry feed formulations. These microalgae not only provide a range of essential minerals but also offer a balanced mineral profile, making them ideal nutritional supplements. The presence of minerals like potassium, iron, magnesium, calcium, iodine, zinc, manganese, and copper in abundant quantities emphasizes their suitability for enhancing poultry nutrition [29,30]. However, the bioavailability and digestibility of these minerals are crucial factors. The cell walls of certain microalgae species can impede the accessibility of these nutrients. Ongoing research aims to overcome this challenge, with techniques such as cell disruption or fermentation being explored to improve the bioavailability of minerals from microalgae in poultry diets [31].

**Table 1 vetsci-11-00044-t001:** Mineral contents and profiles of the main microalgae used in animal feed (dry-matter basis).

Analysis	*Arthrospira* sp.	*Chlorella* sp.	*Isochrysis* sp.	*Porphyridium* sp.	*Schizochytrium* sp.
Ash (%)	6.10–34.8 (9.87 ± 6.00)	5.5–27.3 (10.7 ± 5.4)	12.0–32.2 (18.7 ± 6.14)	16.5–35.9 (23.1 ± 7.62)	3.81–10.0 (7.37 ± 2.35)
Macrominerals (g/kg)					
Ca	0.23–10.3 (3.45 ± 3.78)	0.36–53.3 (9.32 ± 16.8)	5.83–11.5 (9.37 ± 3.08)	6.40–20.7 (12.8 ± 5.17)	3.53
K	10.9–29.1 (18.1 ± 5.84)	0.01–133 (23.6 ± 41.6)	4.10–13.1 (10.4 ± 4.22)	6.70–13.5 (11.2 ± 2.69)	5.71
Mg	0.77–4.00 (2.72 ± 1.20)	0.41–16.4 (5.56 ± 5.69)	3.38–10.0 (6.07 ± 3.03)	4.74–13.7 (7.41 ± 3.61)	NA
Na	4.80–96.2 (25.8 ± 26.0)	0.07–16.5 (5.67 ± 6.81)	11.1–27.4 (18.4 ± 8.26)	8.10–70.7 (29.5 ± 27.4)	1.04
P	1.50–14.8 (9.10 ± 4.25)	5.11–27.1 (16.4 ± 7.37)	6.25–27.6 (15.5 ± 11.0)	3.17–14.6 (10.5 ± 6.39)	4.88
S	NA	0.12	NA	6.40–14.8 (11.9 ± 4.76)	7.68
Microminerals (mg/kg)					
Cu	0.40–18.7 (4.32 ± 6.54)	0.00–119 (24.3 ± 35.4)	6.00–28.0 (14.5 ± 9.75)	7.86–45.3 (17.0 ± 15.9)	2.08
Fe	106–1036 (512 ± 357)	187–5400 (1289 ± 1702)	15.2–2284 (880 ± 1007)	377–11,101 (2682 ± 4708)	13.5
Mn	13.0–550 (87.1 ± 174)	20.9–1270 (269 ± 406)	36.0–834 (272 ± 379)	22.0–259 (81.1 ± 100)	NA
Zn	0.40–30.1 (16.2 ± 11.4)	9.07–530 (131 ± 173)	20.0–940 (280 ± 443)	41.0–392 (199 ± 176)	37.4

Supporting literature: Wild et al. [17], MišurCoVá et al. [20], Altmann et al. [32], Aouir et al. [33], Assaye et al. [34], Assunção et al. [35], Batista et al. [36], Batista et al. [37], Bélanger et al. [38], Bensehaila et al. [39], Bertoldi et al. [40], Cabrita et al. [41], Cabrol et al. [42], Cerri et al. [43], Coelho et al. [44], Coelho et al. [45], Dalle Zotte et al. [46], Di Lena et al. [47], Ferreira et al. [48], Fidalgo et al. [49], Fuentes et al. [50], Fuentes et al. [51], Gamboa-Delgado et al. [52], Habte-Tsion et al. [53], Hadley et al. [54], Holman et al. [55], Holman and Malau-Aduli [56], Karapanagiotidis et al. [57], Kousoulaki et al. [58], Ludevese-Pascual et al. [59], Macias-Sancho et al. [60], Madhubalaji et al. [61], Martins et al. [62], Michael et al. [63], Neylan et al. [64], Oliveira et al. [65], Panahi et al. [66], Prabakaran et al. [67], Radhakrishnan et al. [68], Rohani-Ghadikolaei et al. [69], Sathyamoorthy and Rajendran [70], Shaban et al. [71], Shabana et al. [72], Shields and Lupatsch [73], Sucu [74], Thomas et al. [75], Tibbetts et al. [76], Tibbetts et al. [77], and Tokuşoglu and Üunal [78]. Hyphenated values are ranges based on several studies, followed by average and standard deviation in brackets. NA—not available.

## 4. Impact of Microalgae on Poultry Performance and Egg and Meat Quality

The influence of the dietary inclusion of microalgae on poultry performance has been studied, with varying results depending on the type of microalgae, the concentration used, and the poultry species. These studies primarily focused on the influence of microalgae on average daily gain (ADG), average daily feed intake (ADFI), the feed conversion ratio (FCR), and overall growth performance, including the mineral composition of the microalgae and its effect on these parameters. Some studies also report the effect of microalgae on poultry egg and meat quality. Indeed, low-to-moderate doses of microalgae may improve the physical, chemical, and nutritional attributes of poultry meat and eggs compared to high doses, which could provoke adverse effects [79].

In studies involving *Arthrospira* sp., the dietary inclusion of 4–8% fed to male chicks for 16 days starting at 21 days of age showed no significant effect on ADG [80]. Similar findings regarding ADG, ADFI, and the FCR were reported in chickens with different levels of *Arthrospira* sp. incorporation, including 0.5–1% for 42 days [81], 1.5–2.5% for 4 weeks [82], and 6–21% for 21 days [83]. However, a contrasting result was observed by Shanmugapriya, et al. [84] in which the dietary inclusion of 1% *A. platensis* fed to one-day-old broiler chicks increased ADG and decreased the FCR. In laying hens, a substitution of 9% of soybean meal with *A. platensis* improved egg quality (albumen index, haugh units, yolk index and colour, shell ratio, thickness, and strength) [85]. This improvement in eggshell weight and thickness may be attributed to minerals present in the microalgae [79]. Moreover, the inclusion of a high dose (15%) of *A. platensis* in broilers’ diet increased the total carotenoids in meat but reduced *n*-3 PUFAs and α-tocopherol [86]. Moreover, the contents of Ca and P in tibia ash increased in broiler chickens fed diets containing 3 and 6% *A. platensis* compared with those fed a diet containing 0% SPA [87]. Also, the richness in amino acids in A. platensis may contribute to an improvement in ADG [87].

*Chlorella* sp., another widely studied microalgae, has shown a consistent increase in ADG and a decrease in the feed-to-gain (F:G) ratio in chickens and ducks across various concentrations and trial durations [88,89,90]. In laying hens, Englmaierová et al. [91] reported that the 1.25% dietary inclusion of *Chlorella* sp. decreased the FCR without affecting feed intake. In ducks, a 0.1–0.2% inclusion of *Chlorella* increased feed intake [90]. In broiler chickens, the dietary incorporation of *C. vulgaris* at 10% led to the accumulation of total carotenoids in meat with a slight increase in 18:3*n*-3 [92].

Studies involving *Porphyridium* sp. indicated that chickens fed 5–10% of this microalga for 10 days exhibited a reduced ADFI without an effect on body weight [93]. Conversely, feeding broiler chicks with 0.1–0.2% *Schizochytrium* JB5 for 35 days had no significant effect on ADG, ADFI, or the FCR [94]. However, Ribeiro et al. [95,96] observed increased ADG and ADFI in broilers aged 21 days fed with 7.4% *Schizochytrium* sp. (DHA-Gold extract), although results for the FCR and carcass yield were inconsistent.

The optimal amount of microalgae in feed varies according to both the type of microalgae and the animal species. For example, the dietary inclusion of *A. platensis* in poultry at high percentages (up to 21%) improved productivity with minor effects on meat quality. In the case of *Schizochytrium* sp., higher dietary percentages were used in poultry (7.4%) compared to pigs, ruminants, and rabbits. The most notable impact of *Schizochytrium* incorporation was an improvement in the fatty acid composition of meat, particularly in increasing *n*-3 long-chain polyunsaturated fatty acid (LC-PUFA) levels, including EPA and DHA. *Chlorella*, on the other hand, was consistently included in feed at lower percentages (up to 1.25%) and was found to benefit growth performance in poultry.

Table 2 summarizes the principal impacts of microalgae on poultry performance and egg and meat quality.

## 5. Sustainability and Environmental Impact

The utilization of microalgae as a feed ingredient in poultry nutrition represents not just a nutritional choice but also a significant step toward environmental sustainability. The cultivation of microalgae is particularly notable for its minimal impact on natural resources and its potential contribution to ecological balance.

One of the most substantial benefits of microalgae cultivation is its low reliance on land and freshwater resources. Microalgae can thrive in environments unsuitable for traditional agriculture, such as brackish water and wastewater. This capability is vital in conserving valuable agricultural land and reducing pressure on increasingly scarce freshwater resources, presenting a sustainable alternative to conventional crop cultivation [97].

Additionally, microalgae possess an inherent ability to sequester carbon dioxide, a crucial feature in the battle against climate change. Through the process of photosynthesis, microalgae incorporate CO_2_ into their biomass, effectively reducing greenhouse gas emissions. This attribute is especially beneficial when microalgae are cultivated using CO_2_ emissions from industrial sources, thereby converting a waste product into a valuable resource and contributing to carbon mitigation efforts [98].

The role of microalgae in bioremediation and pollution control is another significant environmental benefit. Microalgae can absorb and utilize nutrients and pollutants from wastewater, purifying the water while simultaneously enriching the microalgae with additional nutrients. This dual benefit makes microalgae not only a sustainable feed component but also a tool for environmental clean-up [99].

Furthermore, the use of microalgae in poultry feed can aid in biodiversity conservation. Traditional feed ingredients, such as fishmeal and soybean meal, are often linked to overfishing and deforestation. Microalgae offer an alternative nutrient source, potentially reducing the exploitation of these natural resources and helping preserve biodiversity [100].

Lastly, the lifecycle environmental impact of microalgae production is generally more favourable compared to conventional feed ingredients. Studies have indicated that the emissions and energy consumption associated with microalgae production are typically lower. However, it is crucial to recognize that certain energy-intensive cultivation methods can lessen these environmental benefits. Ongoing research and technological advancements are therefore essential to optimizing microalgae production methods to ensure they are as environmentally friendly as possible [101]. The replacement of conventional ingredients in monogastric animals´ diets with microalgae would possibly contribute to reduce some environmental adverse effects, like pesticides and greenhouse gas emissions [102].

Table 3 summarizes the main impacts of microalgae on sustainability and environment.

## 6. Economic Viability

The economic viability of the integration of microalgae into poultry diets is a critical factor that influences its practicality for widespread commercial adoption. Assessing the costs associated with microalgae production, processing, and incorporation into poultry feed, particularly in comparison with traditional feed ingredients, is essential to understanding its feasibility.

The primary factor contributing to the cost of using microalgae in poultry feed is the cultivation process. Microalgae can grow rapidly in a variety of conditions, but establishing and maintaining cultivation systems, such as open ponds and photobioreactors, can be costly. Open ponds are more cost-effective but often face challenges with contamination control and consistent yields. On the other hand, photobioreactors provide better control over growing conditions but are more capital-intensive to set up and maintain [103].

Another significant contributor to cost is the harvesting and processing of microalgae. Due to the small size and low density of microalgae cells, harvesting methods like centrifugation, filtration, and flocculation can be energy-intensive and thus expensive. Additionally, further processing steps such as cell disruption, which are often necessary to increase the bioavailability of nutrients in microalgae, add to the overall production costs [104].

The cost-effectiveness of microalgae is also influenced by the bioavailability of their nutrients in poultry. If these nutrients are not readily available, it may necessitate higher inclusion rates in feed, thereby increasing cost [11]. This factor is crucial when considering the economic feasibility of microalgae as a feed alternative.

When comparing the costs of microalgae to traditional feed ingredients like soybean meal and fishmeal, it is important to note that microalgae must be competitive in terms of both nutritional content and cost. While microalgae provide superior nutritional benefits, studies have indicated that the cost per unit of protein or essential minerals in microalgae is currently higher than in traditional feed sources [105].

Furthermore, the potential market for microalgae-based poultry feed is influenced by factors such as consumer willingness to pay for poultry products with enhanced nutritional profiles and regulatory incentives for sustainable agricultural practices. The unique health benefits offered by microalgae, like the omega-3 fatty acid enrichment in eggs, can create niche markets where consumers are willing to pay a premium for these enhanced products [11].

Table 4 summarizes the principal aspects of microalgae with respect to economic viability.

## 7. Safety and Regulatory Aspects

The inclusion of microalgae in poultry diets brings forth a need for a comprehensive evaluation of safety and adherence to regulatory standards. This assessment is crucial in understanding the potential health risks associated with the use of microalgae in animal feed and ensuring compliance with the regulatory framework that governs their application.

A primary safety concern with microalgae is the risk of toxin and contaminant presence. Some species of microalgae, especially those cultivated in open ponds, are prone to contamination by heavy metals and other environmental pollutants. In addition, certain microalgae are capable of producing toxins, such as microcystins, which pose health risks not only to poultry but also to human consumers if these contaminants enter the food chain. Regular monitoring and stringent quality control measures are vital in mitigating these risks to maintain the safety of poultry feed and, consequently, poultry products [106].

Another aspect to consider is the potential for allergenic reactions to components of microalgae. Though this area is relatively unexplored, it is an important consideration given the novel nature of microalgae as a feed ingredient. Additionally, the digestibility of microalgae is a crucial factor in ensuring the safety and effectiveness of their nutrients for poultry. Factors like the composition of microalgal cell walls can influence digestibility, impacting the availability of nutrients to poultry [107].

The regulatory frameworks governing the use of microalgae in animal feed vary across regions. In the European Union, microalgae intended for animal feed must comply with regulations set by the European Food Safety Authority (EFSA), encompassing safety, efficacy, and environmental impact assessments. Similarly, in the United States, the Food and Drug Administration (FDA) is responsible for overseeing the approval of new feed ingredients under the Food, Drug, and Cosmetic Act, ensuring their safety and effectiveness [108].

Furthermore, transparency in labeling and consumer information is crucial, especially for novel feed ingredients like microalgae. The accurate labeling of poultry products that are derived from birds fed microalgae-based diets is essential. Such labeling should inform consumers about any specific health benefits or changes in the nutritional profile of these products [109].

In conclusion, while microalgae present a promising alternative to conventional poultry feed ingredients, ensuring their safety and regulatory compliance is paramount. Rigorous testing, adherence to quality control measures, and alignment with regulatory standards are essential steps in establishing microalgae as safe and effective feed ingredients in the poultry industry.

Table 5 summarizes the principal impacts of microalgae on sustainability and the environment.

## 8. Conclusion and Future Perspectives

The exploration of microalgae as a novel and sustainable source of minerals for poultry feeding has revealed a landscape rich with potential and challenges. Microalgae emerge as a powerhouse of nutrition, offering a blend of essential minerals, proteins, vitamins, and bioactive compounds. Their incorporation into poultry diets promises not only to enhance the nutritional quality of poultry feed but also to improve the health and productivity of the birds. Significantly, the environmental sustainability of microalgae cultivation is a notable advantage. With their minimal land and water requirements, capacity for carbon sequestration, and bioremediation potential, microalgae present a solution aligned with the goals of sustainable agriculture and environmental stewardship.

However, the journey from potential to practice in the use of microalgae in poultry feed is not without hurdles. The economic aspect, primarily the cost associated with their cultivation and processing, stands as a significant barrier to the widespread adoption of microalgae in commercial poultry diets. This economic challenge underscores the need for continued innovation and research in optimizing cultivation methods and reducing overall production costs. Furthermore, the safety of microalgae as a feed ingredient, particularly concerning the presence of toxins and heavy metals, requires rigorous assessment and monitoring. Ensuring compliance with stringent regulatory standards and maintaining transparency with consumers through proper labeling and communication are also crucial for gaining public trust and acceptance.

Looking ahead, the future of microalgae in poultry nutrition holds promise but is dependent on overcoming these economic and safety challenges. Advancements in biotechnology, cultivation techniques, and processing methods are essential to enhancing the feasibility and reliability of microalgae as a feed ingredient. As the world increasingly looks toward sustainable solutions in agriculture, microalgae have the potential to play a transformative role in poultry nutrition. Embracing this potential will not only contribute to more sustainable poultry production but also resonate with the broader objectives of global food security and environmental conservation.

## Figures and Tables

**Table 2 vetsci-11-00044-t002:** Summary of principle impacts of microalgae on poultry performance and egg and meat quality.

Microalga	Dietary Inclusion and Animals	Impact	Reference
*Arthrospira* sp.	4–8% in 21-day-old male chicks for 16 days	No significant effect on ADG	[80]
	1% in 1-day-old broiler chicks	Increased ADG and decreased FCR	[84]
	9% in laying hens	Improved egg quality	[85]
	15% in 1-day-old broilers	Increased total carotenoids in meat, but reduced *n*-3 PUFA and α-tocopherol contents	[86]
	3 and 6% in broilers	Increased Ca and P in tibia ash	[87]
*Chlorella* sp.	1.25% in laying hens	Decreased FCR without affecting feed intake	[91]
	10% in 1-day-old broilers for 21 days	Increased total carotenoids and a small significant increase in 18:3*n*-3	[92]
	15% in 1-day-old broilers for 21 days	Increased total carotenoids but decreased *n*-3 PUFA and α-tocopherol contents	
	0.1–0.2% in 1 -day-old ducks	Increased feed intake	[90]
*Porphyridium* sp.	5–10% in 12–13, 30-week-old chickens	Reduced ADFI without affecting body weight	[93]
*Schizochytrium* JB5	0.1–0.2% in 2-day-old broilers for 35 days	No significant effect on ADG, ADFI, or FCR	[94]
*Schizochytrium* sp.	7.4% in 21-day-old broilers	Increased ADG and ADFI	[95,96]

ADFI, average daily feed intake; ADG, average daily gain; FCR, feed conversion ratio, PUFA, polyunsaturated fatty acid.

**Table 3 vetsci-11-00044-t003:** Summary of the main impacts of microalgae on sustainability and environment.

Impacts	Reference
Low reliance on land and freshwater resources for cultivation; Microalgae thrive in environments unsuitable for traditional agriculture.	[97]
Microalgae sequester carbon dioxide, a crucial feature in the battle against climate change. Through the process of photosynthesis, microalgae incorporate CO_2_ into their biomass, effectively reducing greenhouse gas emissions.	[98]
Microalgae assist in bioremediation and pollution control.	[99]
Microalgae are an alternative nutrient source, potentially reducing the exploitation of natural resources and helping preserve biodiversity.	[100]

**Table 4 vetsci-11-00044-t004:** Summary of the economic viability aspects of microalgae in poultry nutrition.

Aspects	Reference
Establishing and maintaining cultivation systems can be costly; open ponds are more cost-effective; photobioreactors provide better control over growing conditions.	[103]
Harvesting and processing can be energy-intensive and thus expensive.	[104]
Cost-effectiveness is influenced by the bioavailability of nutrients in poultry.	[11]
Microalgae must be competitive in terms of both nutritional content and cost.	[105]

**Table 5 vetsci-11-00044-t005:** Summary of safety and regulatory aspects of microalgae for poultry nutrition.

Aspects	Reference
Risk of toxin and contaminant presence; Capable of producing toxins such as microcystins; Regular monitoring and stringent quality control measures are vital.	[106]
The potential for allergenic reactions to components of microalgae, an important consideration given the novel nature of microalgae as a feed ingredient.	[107]
Regulatory frameworks governing the use of microalgae in animal feed vary across regions. In the European Union, microalgae intended for animal feed must comply with regulations set by the European Food Safety Authority (EFSA), and in the United States, the Food and Drug Administration (FDA) is responsible for overseeing the approval of new feed ingredients.	[108]
Transparency in labeling and consumer information is crucial, especially for novel feed ingredients like microalgae.	[109]

## Data Availability

The data presented in this study are available upon request from the corresponding author.

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
