# Peer review of "Microalgae as an Alternative Mineral Source in Poultry Nutrition"

_vetsci, 2024, doi:10.3390/vetsci11010044_

Round 1

Reviewer 1 Report

Comments and Suggestions for Authors

The present review article deals with a very relevant topic, regarding microalgae as a sustainable and nutritionally rich alternative for mineral supplementation in poultry diets.

Microalgae are recognized as a great source of omega-3 polyunsaturated fatty acids, as well as essential elements (like selenium, iodine, etc), and numerous bioactive compounds. For this reason, the incorporation of microalgae into poultry diets could present numerous advantages for chickens’ health along with improved poultry meat and eggs quality, while reducing environmental footprint associated with diets production. Based on this, authors address in this article, multiple positive and negative issues to be considered.

This interesting article is well written, and its conception included very pertinent topics, yet some minor issues should be addressed. They are listed below:

1.     In lines 79 and 92 authors designate omega-3 fatty acids in different manners. These designations should be harmonized;

2.     In line 118 the expression “in vitro” should be formatted as italic;

3.     In Table 1, thousands are not separated by comma. The same should be kept on the text (for example in lines 149, 154 and 159);

4.     Despite of the impact of microalgae on poultry performance is well explained, it would be interesting to include some considerations about possible improvements on meat or eggs quality as well.

5.     Line 213 and 227: replace Arthrospira palentesis by A. palentesis and format in italic.

6.     Line 216: please explain the abbreviation F:G;

7.     Line 247: format the “2” in CO2 as subscript;

8.     Regarding the considerations included in sections 5, 6 and 7 it would be interesting that they could be better supported using relevant data. For example:

a.     Is there any available data that could be used to compare the environmental foodprint of conventional diets vs microalgae diets? (Section 5)

b.     Are there any comparative studies discussing the costs for the production conventional diets vs microalgae diets? (Section 6)

Author Response

Reviewer 1

The present review article deals with a very relevant topic, regarding microalgae as a sustainable and nutritionally rich alternative for mineral supplementation in poultry diets.

Microalgae are recognized as a great source of omega-3 polyunsaturated fatty acids, as well as essential elements (like selenium, iodine, etc), and numerous bioactive compounds. For this reason, the incorporation of microalgae into poultry diets could present numerous advantages for chickens’ health along with improved poultry meat and eggs quality, while reducing environmental footprint associated with diets production. Based on this, authors address in this article, multiple positive and negative issues to be considered.

This interesting article is well written, and its conception included very pertinent topics, yet some minor issues should be addressed. They are listed below:

Reply: Thank you for your comments and suggestions. We appreciate it. We tried to address all of them.

  1. In lines 79 and 92 authors designate omega-3 fatty acids in different manners. These designations should be harmonized;

Reply: Thank you for the suggestion. The designation was harmonized.

  1. In line 118 the expression “in vitro” should be formatted as italic;

Reply: Thank you for the suggestion. Formatted.

  1. In Table 1, thousands are not separated by comma. The same should be kept on the text (for example in lines 149, 154 and 159);

Reply: Thank you for the suggestion. Changed.

  1. Despite of the impact of microalgae on poultry performance is well explained, it would be interesting to include some considerations about possible improvements on meat or eggs quality as well.

Reply: Thank you for the suggestion. Some considerations about possible improvements on meat or eggs quality are now present in the reviewed manuscript, section 4, lines 207-210, 217-223, page 7, and in Table 2.

  1. Line 213 and 227: replace Arthrospira palentesis by A. palentesis and format in italic.

Reply: Thank you for the suggestion. Changed.

  1. Line 216: please explain the abbreviation F:G;

Reply: Thank you for the suggestion. Explained.

  1. Line 247: format the “2” in CO2 as subscript;

 Reply: Thank you for the suggestion. Formatted.

  1. Regarding the considerations included in sections 5, 6 and 7 it would be interesting that they could be better supported using relevant data. For example:
  2. Is there any available data that could be used to compare the environmental foodprint of conventional diets vs microalgae diets? (Section 5)

 Reply: Thank you for the suggestion. A paragraph concerning the environmental foodprint of conventional diets vs microalgae diets is present in the reviewed manuscript, section 5, lines 274-276, page 8 and in Table 3.

  1. Are there any comparative studies discussing the costs for the production conventional diets vs microalgae diets? (Section 6)

 Reply: Thank you for the suggestion. A paragraph concerning the costs for the production of conventional diets vs microalgae diets is present in the manuscript, section 6, lines 299-303, page 8.

Reviewer 2 Report

Comments and Suggestions for Authors

Dear authors, the review looks interesting. Good luck with your work.

Author Response

Reviewer 2

Dear authors, the review looks interesting. Good luck with your work.

 Reply: Thank you for your comments and suggestions. We appreciate it.

Reviewer 3 Report

Comments and Suggestions for Authors

Very interesting review, well written and developed. 

I did not found - although I have not verified the literature in detail - mistakes or incorrect information. But I am a little disappointed - as a poultry nutritionist - that no information is presented concerning e.g. egg shell quality or bone development in chapter 4. The information concerning ADG ... FCR is interesting as e.g. protein and amino acid  source. However microalgae will primarily be used as a mineral source as stated in your summary / introduction and conclusion. If this information is lacking in the literature, authors have to mention this clearly.

Another remark concerning the costs (L293-294) actually higher than traditionally sources. Is this 2 times or 20 times ? 

A suggestion. Only one (but very interesting) table in this review but lots of text. However, this makes it not very readable. With a picture or is it not possible to insert an overview table in which the positive points, the risk factors and the potentials are summarized. The paper will be more attractive and in less time be readable. Only very, very few people read still pages without pictures, quotes, photo's etc.... Even for a scientific paper, the readability is important. 

However, this is a suggestion not a claim from my side.

Author Response

Reviewer 3

Very interesting review, well written and developed. 

I did not found - although I have not verified the literature in detail - mistakes or incorrect information. But I am a little disappointed - as a poultry nutritionist - that no information is presented concerning e.g. egg shell quality or bone development in chapter 4.

Reply: Thank you for your comments and suggestions. We appreciate it. We tried to address all of them. A paragraph concerning eggshell quality is present in the reviewed manuscript, section 4, lines 207-210 and 2017-223, page 6, and in Table 2 Also bone development, section 4, lines 222-224.

The information concerning ADG ... FCR is interesting as e.g. protein and amino acid  source. However microalgae will primarily be used as a mineral source as stated in your summary / introduction and conclusion. If this information is lacking in the literature, authors have to mention this clearly.

Reply:  Thank you for your comments and suggestions. A paragraph concerning the importance of amino acids is presented in the reviewed manuscript, section 4, lines 224-225.

Another remark concerning the costs (L293-294) actually higher than traditionally sources. Is this 2 times or 20 times? 

Reply: Thank you for your comment. According to the literature, currently, the direct costs of microalgae compare poorly with traditional protein. Soy costs less than US$1 per kilogram to produce and beef costs approximately US$3 per kilogram, but microalgae is US$10 per kilogram (Nature Portefolio, Steak expectations for alternative protein sources, 2021).

A suggestion. Only one (but very interesting) table in this review but lots of text. However, this makes it not very readable. With a picture or is it not possible to insert an overview table in which the positive points, the risk factors and the potentials are summarized. The paper will be more attractive and in less time be readable. Only very, very few people read still pages without pictures, quotes, photo's etc.... Even for a scientific paper, the readability is important. 

However, this is a suggestion not a claim from my side.

Reply: Thank you for your suggestion. We incorporated Tables 2, 3, 4 and 5 to summarize the aspects in each section. in the reviewed manuscript.